# Targeted Telehealth Education Increases Interest in Using Telehealth among a Diverse Group of Low-Income Older Adults

**DOI:** 10.3390/ijerph192013349

**Published:** 2022-10-16

**Authors:** Emily Jezewski, Abigale Miller, MaryAnn Eusebio, Jane Potter

**Affiliations:** 1College of Medicine, University of Nebraska Medical Center, Omaha, NE 68198, USA; 2Eastern Nebraska Office on Aging, Omaha, NE 68137, USA; 3Division of Geriatrics, Gerontology, and Palliative Medicine, Department of Internal Medicine, University of Nebraska Medical Center, Omaha, NE 68198, USA

**Keywords:** telehealth, outreach, health promotion, older adults, COVID-19

## Abstract

Telehealth allows older adults to take control over their health and preventive care; however, they are less likely to use telehealth. Minority older adults use telehealth services less than their White counterparts. During COVID-19, the U.S. Medicare system allowed for telehealth delivery of Annual Wellness Visits, which are known to improve use of preventive services. To increase telehealth use, we targeted vulnerable, low-income, minority older adults and provided education to improve knowledge of and identify barriers to telehealth use. Ultimately, this could serve as a means of improving health and preventive care services. Participants resided at independent living facilities, low-income housing, and elders of the Native American coalition; N = 257. Participants received written education materials; a subset attended a 20-min presentation. In this quasi-experimental study, participants completed a pre-post survey. Results were analyzed using Chi-Squared and Fisher’s Exact tests. Participants included 54 ‘in-person’ and 203 ‘at-home’ learners. Most were female (79%), single/widowed (51%), and white (65%). At baseline, 39% were familiar with telehealth; following education 73% stated understanding on accessing telehealth. Nearly 40% of participants said they would use telehealth in the future; a larger proportion of “in-person” (73%) learners were willing to use telehealth than “at-home” learners (41%) (*p* = 0.001). Divorced older adults and Blacks voiced greater likelihoods of using telehealth than their married/widowed and White counterparts, respectively (Χ2(3, N = 195) = 9.693, *p* = 0.02), (*p* = 0.01). This education program demonstrates an increase likelihood in health promotion among older adults by increasing confidence in accessing and future use of telehealth; therefore, we achieved our aim of promoting telehealth use and improving health promotion.

## 1. Introduction

Studies suggest 3.6 million adults in the United States (U.S.) are homebound due to underlying functional impairments [1,2,3]. As a result, it is challenging for this population to obtain office-based healthcare and wellness services [4]. Access to office visits is further reduced among older adults who have less social support [4]. This can lead to an increased number of emergency room visits and hospitalizations from preventable illness. Altogether these consequences result in increased healthcare costs and increased mortality among older adults [5].

The World Health Organization defines health promotion as: “the process of enabling people to increase control over, and to improve their health” [6]. The Medicare Annual Wellness visits (AWV), introduced in the US in 2011, improved access to and increased utilization of preventive services by eliminating cost to beneficiaries [7]. Data shows that the use of AWVs leads to an increased use of preventive services, and overall health promotion [7]. Yet, there are disparities in who accesses this benefit. For example, non-Hispanic black and non-Hispanic other races are far less likely to complete an AWV [7]. Older adults also sometimes face barriers to health promotion from limited mobility and healthcare access. Several solutions have been proposed to mitigate these barriers. These include electronic measuring tools, such as phone applications or online health tool monitoring, as well as direct nursing interventions [8,9]. While interventions targeted at increasing health promotion can be difficult to implement, studies suggest that overall, these types of interventions lead to lower healthcare costs, improvements in quality of life and overall well-being [10]. We propose that one newer intervention aimed to increase health promotion in older adults, particularly in the COVID-19 Era, is telehealth. Telehealth may be a worldwide solution for older adults to help take control of their healthcare and wellness from their own homes. In the U.S., the public health emergency brought on by the pandemic allowed for the virtual delivery of AWV; whether this benefit will continue beyond the public health emergency is uncertain. 

Telehealth services truly altered healthcare delivery during COVID-19 in the U.S. [11]. Among Medicare beneficiaries specifically, there was a 63-fold increase in telehealth use from 840,000 visits in 2019 to 52.7 million in 2020 [10]. However, telehealth use differs by race, ethnicity, location, age, and insurance payer with lower rates among older adults, Blacks, rural residents, and individuals with Medicare/Medicaid [12,13,14,15,16,17,18,19]. 

Among older adults, estimates suggest about 38% of this population is not prepared to participate in video visits and 20% are unable to participate in phone visits [20]. One barrier to video visits is reliable internet access, which is a significant barrier for older adults in the U.S. [21]. For example, in New York City in 2017, nearly half of adults over 65 lacked access to the internet, compared to only ~20% of younger adults [22]. Trends in internet access are related to both poverty and location [13]. Additional barriers to telehealth cited specifically for older adults include cost, inexperience, lower confidence in using telehealth, and lack of help from a caregiver [15,21]; as well as age-related barriers, such as hearing or vision impairment, mild cognitive impairment, or dementia [20]. Despite the significant barriers, studies show that education and health promotion interventions can increase telehealth use among older adults [21]. Important to note are the numerous benefits to telehealth, including less deferred care, reduced travel barriers, improved communication with caregivers, and improved patient wellbeing [23]. 

Other projects demonstrate that education increases use of and confidence in using telehealth [24]. With COVID-19 and the associated changes to healthcare delivery, we found it exceptionally important to address telehealth via education to keep older adults safe at home and with the ability to access healthcare. Older adults are at increased risk of severe infection and morbidity from COVID-19 [25,26,27]. As we move slowly beyond the COVID-19 Pandemic, telehealth will remain vital for older adults’ health and wellness. Thus, the goal of this project was two-fold. The first goal was to reduce telehealth barriers in vulnerable people through education and examine residual barriers/facilitators post-education. Secondarily, we hoped to improve older adults’ access to healthcare to increase health promotion and preventive services among a vulnerable population.

## 2. Materials and Methods

This quasi-experimental study used a pre-post questionnaire to evaluate the effectiveness of an educational intervention to improve knowledge of and self-reported likelihood of use of telehealth services. The IRB at The University of Nebraska Medical Center deemed this project as exempt, as it was not human-subjects research. The IRB stated this qualified as a service-learning and quality improvement project. Participant identifiers, such as names, were not collected and participants were informed of this. Consent was implied when participants chose to complete a survey.

We partnered with our local Area Agency on Aging (AAA). This is an agency created and funded by the U.S and State governments, which publicizes itself and as a “one-stop shop” for programming, services, and housing options for older adults in different communities. With our local AAA, we developed telehealth education for low-income older adults in residential and community settings with a goal to reach 600+ individuals. In doing so, we hoped to empower older adults to take control of their healthcare through telehealth. 

To identify older adult participants, we contacted 6 living facilities, 5 community groups, and Douglas County Housing Authority Properties. These housing properties were selected because they follow the Department of Housing and Urban Development and Low-Income Tax Credit guidelines. Additionally, a presentation was delivered to the Native American Coalition. 

The education included written guides and a 20-min presentation detailing telehealth. We employed a pre-post-survey to assess the effect of the educational intervention. The original plan was to administer the pre-survey and then deliver the oral presentation with a built-in question/answer session and provide participants with the written guides. Following this we planned to administer the post-survey. Educational interventions took place during the Summer and Fall of 2020; given this timeframe, there were significant COVID-19 precautions in place, which limited our ability to schedule in-person education sessions. As a result, we were not able to carry out our original plan and only delivered 6 presentations ‘in-person’. 

For the groups who could not receive an ‘in-person’ presentation due to COVID restrictions, we provided them with the written guides and a paper version of the oral presentations for review ‘at-home’. ‘At home’ participants received a survey as well, with instructions to complete the pre-survey before going through educational materials and to complete the post-survey following the education. 

We also originally planned follow-up one-on-one meetings to demonstrate telehealth on personal devices, which was also limited by public health precautions. 

The surveys were designed to capture demographics (age, race) and other characteristics (available internet and devices) thought to influence telehealth use. The pre-survey, completed before education, collected: age, race, living setting, marital status, gender, and access to telehealth devices (phone, laptop/tablet). The pre-survey included yes/no questions: “Have you avoided the doctor because of COVID-19?” “Are you familiar with telehealth?” “Do you have access to the internet?” 

The post-survey, completed after education, asked the following: “Do you have someone to help you with telehealth?” “Do you have a better understanding of telehealth?” “After reviewing the materials, were all of your questions answered?” “Would you like more information about telehealth?” “Would you use telehealth?” 

Variables were analyzed separately for those who learned ‘in-person’ versus ‘at home’ in order to examine relative effectiveness by site of education and demographics. Yes/no responses were analyzed by gender, race, age, and marital status. Race was split into White/Black; gender was split into male/female. Race/gender analyses used Fisher’s Exact test. Age groups were: <65, 66–75, 76–85, >85; marital status: married, divorced, widowed, or single. Yes/no questions analyzed by age/marital status were completed using Chi-Squared test with Fisher’s exact test for post hoc. 

All fully completed surveys were used in analysis. For any given analysis, incomplete responses were left out of calculations. Overall, the presentation and written guides, along with the surveys made up an education “packet.” A total of 630 education packets were given to older adults. Fifty-four (54) of these were to participants who attended ‘in-person’ presentations; the remaining 576 were delivered to ‘at-home’ participants. All ‘in-person’ attendees completed the surveys. Of the ‘at-home’ participants, only returned surveys (203/576) were used in analysis. 

All materials used in this project are available at: www.unmc.edu/NebraskaGWEP/public-education/telehealth-contacting-your-provider-via-phone-computer/ (accessed on 26 July 2022). Analyses used MedCalc statistical software. A *p*-value < 0.05 was considered significant. The Health Resources and Services Administration partially funded this project, grant: T1MHP390775. 

## 3. Results

We completed 6 ‘in person’ presentations with distribution of written materials to 7 other groups, reaching a total of 630 older adults; 54 attended ‘in-person’ and completed the survey. Two-hundred-three (203) ‘at-home’ learners returned surveys by mail; see Table 1. There were differences in race (more Black, *p* < 0.001) and age (older, Χ2(3, N = 250) = 16.581, *p* = 0.009), but not gender or marital status among ‘in-person’ versus ‘at-home’ learners.

On the pre-survey (baseline), 17% of respondents reported avoiding medical encounters due to COVID-19. On the post-survey, 93% reported access to a telehealth-compatible device and 41% reported there was someone who could assist them. Of the ‘at-home’ learners, 4% did not have access to a telehealth compatible device, compared to 14% of ‘in-person’ learners. At baseline, 36% were familiar with telehealth; after education, 70% understood how to access telehealth and 39% stated they would use it. There was no difference in understanding telehealth access between ‘in-person’ and ‘at-home’ learners (Χ2(1, N = 224) = 2.585, *p* = 0.108). 

After education, 21% wanted more information. This differed between ‘in-person’ (52%) versus 13% of ‘at-home’ (Χ2 (2, N = 225) = 39.863, *p* < 0.001). Likelihood of future use was 73% for ‘in-person’ and 41% for ‘at-home’ (*p* = 0.001); future use was greater in Blacks versus Whites for all participants (*p* = 0.01); this difference remained true only for the ‘at-home’ group when groups were examined separately (*p* = 0.008). See Figure 1. 

Blacks were more likely to request more information in ‘in-person’ and ‘at-home’ populations (*p* = 0.009, *p* < 0.001). Divorced participants reported a higher likelihood of future use (Χ2(3, N = 195) = 9.693, *p* = 0.02) than married and widowed participants. This was true for the entire population (N = 257) and ‘at-home’ learners (Χ2(3, N = 162) = 18.551, *p* = 0.0003). Age and gender did not influence requests for more information or future use. 

## 4. Discussion

We were interested in increasing telehealth understanding and improving access to health and preventive services among minority and low-income older adults. To that end, we distributed education materials on telehealth to 630 low-income older adults and analyzed the effect of this education among 257 survey respondents. In those completing the surveys, familiarity with telehealth increased from 36% to 70%, with ~40% saying they would use telehealth. Of these, most received only written materials for ‘at-home’ learning (N = 576). Of the ‘at-home’ learners, 35% returned a completed pre-post survey, for a total of 203 ‘at-home’ participants. Simply distributing materials resulted in educating more than 200 individuals on the benefits of telehealth and how to use it in the comfort of their own homes. There may be additional ‘at home’ learners who did not complete and return our survey who gained knowledge of telehealth. Importantly, the ‘at-home’ group requested additional information less than those attending ‘in-person,’ perhaps because they reviewed materials at their own pace rather than through a brief presentation. However, more ‘in-person’ learners reported future telehealth use, suggesting in-person education resulted in greater confidence with telehealth. 

Many social determinants of health, which include social, economic, and environmental conditions, impact a person’s ability to stay safe and healthy at home and have access to high-quality healthcare with preventive services [28,29]. One social determinant that is key to recognizing in the context of our project includes access to a network providing social support. For example, we demonstrated that divorced older adults voiced greater likelihood to use telehealth, especially among ‘at-home’ learners. Divorced people may have less available transportation to appointments due to limited social networks or strained finances, and telehealth could reduce this barrier. We know that transportation is a limitation for many older adults. In fact, studies demonstrate that on average older adults drive less than their younger counterparts and travel shorter distances [30,31]. We see increased transportation disparities among late-life immigrants and those with language or cultural barriers as well [30]. Limited transportation is also linked with social isolation, which is significantly associated with poorer health outcomes [30,32]. Transportation will be a barrier to healthcare beyond the Pandemic and telehealth may be part of the solution. 

Another key social determinant is income and the ability to afford telehealth compatible devices [33,34,35]. Smartphones, computers, and laptops are costly; therefore, device access/ownership is a direct barrier for telehealth among older adults of lower socioeconomic status. Similarly, if an individual cannot afford a phone or internet access, they cannot use telehealth. Data also suggests that the number of older adults with reliable internet access is low [22]. This is not only important to understand in the context of this project, but also necessary to know that lower socioeconomic status has been associated with accelerated aging [29]. 

Interestingly, this project identified that 93% of participants had a telehealth compatible device, including a telephone, cellphone, smartphone, laptop, or tablet. Previous studies in Medicare beneficiaries identified only ~40% as having access to a laptop or smartphone with internet connection [20]. This difference is likely due to a bias in our sample, where those returning surveys did so because they had devices compatible for telehealth. Notably, more individuals among the ‘in-person’ group lacked access to a personal device capable of telehealth; given that these individuals mostly resided in low-income housing suggests that they may have more limited finances compared to those in the ‘at home’ group. 

Additionally, Black participants were more likely to request additional information compared to Whites. Perhaps Black participants were aware of their greater pandemic-associated risk and had greater interest; or our education was not optimal for this audience. One-on-one sessions might have addressed this deficit. Similarly to above, most of our Black participants were at ‘in-person’ sessions and therefore resided at low-income housing. This could suggest that this population of Black older adults had fewer resources and could not afford telehealth devices, highlighting a potential problem with some populations struggling with device acquisition. As such, this may help to explain why our Black participants requested more information, possibly wondering how to obtain devices or needing overall more assistance with higher-order tasks.

A study, similar to ours, also suggests that in-person education can identify and address telehealth barriers, and help older adults overcome these. This specific study from 2020 took place in patient’s homes; it identified individual barriers, categorized older adults by specific need and provided education addressing those barriers. Thirty—two home telehealth education visits were conducted through which individual barriers to access telehealth were addressed. Participants reported improved well-being with these visits [24]. Importantly, our intervention suggests that you do not need to identify specific barriers, but rather providing generalized education can increase interest in, understanding of and intent to use telehealth in the future in an older, low-income, and diverse population especially through in-person education.

It is important to note that there was a large difference between the number of ‘in-person’ versus ‘at-home’ learners. While a greater proportion of ‘in-person’ learners stated they would use telehealth in the future, compared to ‘at-home’ learners, this conclusion may be difficult to apply to the general older adult population. More ‘in-person’ interventions are needed to determine the strength of this finding. 

While this study was designed for and conducted in the U.S., there are lessons for the older populations residing in other countries. During the COVID-19 Pandemic, interest in and use of telehealth increased worldwide, mostly in higher-income countries, such as in the United States of America, the United Kingdome, Italy, India, Canada and Australia [36]. There is data to suggest similar barriers to telehealth exist outside of the U.S., such as cost, resistance to change and challenges with reimbursement [37]. Specific examples cited include educational level in Belgium, computer literacy level in the Netherlands, and resistance to change in Australia [37]. Globally, the popular sentiment is that telehealth can benefit older adults and barriers specific to older adults are similar to our identified barriers: need for increased older adults and caregiver use, inexperience with technology, and availability [38]. Given the similarity of challenges facing people worldwide, education and outreach interventions such as this one might benefit individuals elsewhere. However, we need to continue to be aware that telehealth services are mostly gaining popularity in high-income countries and that their use remains lower in low-income countries, which will need to be addressed if telehealth is to reduce health disparities and improve delivery of preventive services in the future [36,38]. 

We chose to implement a pre-post test design. Benefits of this design include simple structure and ease of implementation. This was beneficial in our project since in-person presentations were limited and directions on how to perform the assessments could be easily conveyed to participants. Our hopes in doing this was to allow participants of different educational backgrounds to participate with little instruction. We also wanted to determine the level of telehealth understanding before the intervention. However, limitations of this design include placebo effects, as well as difficulty determining that the increased in understanding was truly due to our intervention. This latter limitation is quite important since the vast majority of participants received education for ‘at-home’ review. Since they had an unstructured amount of time to complete and return the survey, they had the advantage of learning more about telehealth on their own before completing and returning the post survey.

Some other limitations of this project include small sample size of ‘in-person’ learners and self-selection of those who returned surveys when receiving only written materials. This self-selection likely explains the high (93%) of survey respondents who reported telehealth ready devices. Further the original plan to hold one-on-one sessions following ‘in-person’ education was not possible due to pandemic-related social distancing. Because the groups of learners (at-home versus in-person) differed in sex, race and age, any comparisons between the two groups must be made with caution. They may be best viewed as two separate populations.

Additionally, since we did not collect contact information of participants, we could not assess their true use of telehealth following our intervention. This data would be useful to have in future projects to better understand the impact of the intervention. Finally, our populations of White and Black participants and age-groups of participants were not equal, in fact there were significant differences between the ‘at-home’ and ‘in-person’ demographics. Therefore, while our conclusions suggest that Black participants were more interested than White, these results may be difficult to generalize, and further education and assessment is needed. Never-the less, this project shows an impact in both raising awareness and improving reported likelihood of future use of telehealth in a diverse and low-income older population. 

## 5. Conclusions

As demonstrated, our intervention successfully showed an increased understanding of and future use of telehealth. Black older adults could benefit from directed education interventions to increase health promotion and telehealth use. Education methods may be useful in the future to increase telehealth use and to increase health promotion activities among vulnerable populations. Overall, technology, including telehealth, may be leveraged to begin closing the gaps in wellness activities created by deficits in social determinants of health and increase the health and well-being in older adults. 

## Figures and Tables

**Figure 1 ijerph-19-13349-f001:**
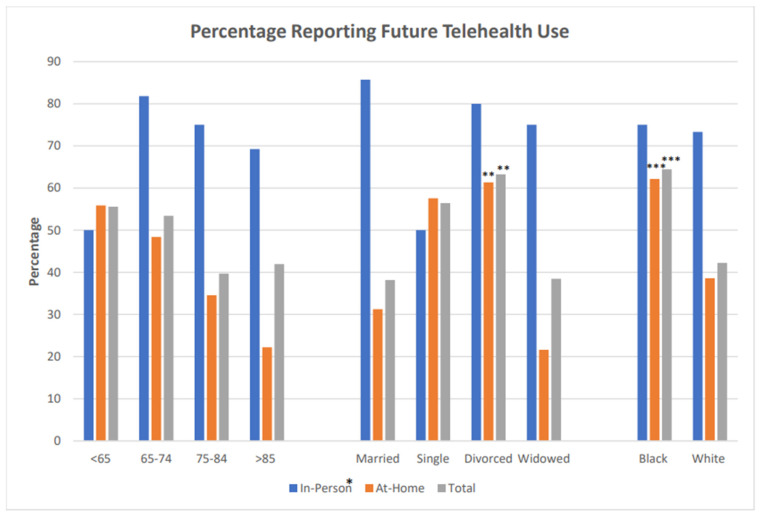
Differences in Stated Likelihood of Future Use. * In-Person learners reported increased likelihood of future use compared to at-home learners. ** Divorced participants were more likely than married or widowed participants to use telehealth among at-home learners and for the entire population. *** Black participants reported increased likelihood of future use compared to white participants.

**Table 1 ijerph-19-13349-t001:** Demographics for Survey Respondents *.

		Total	In-Person	At-Home
Race			N (%)	N (%)
	White	165 (65)	19 (35)	147 (72)
	Black	64 (25)	22 (41) **	42 (21)
	Native American	6 (2)	5 (9)	1 (<1)
	Other	3 (1)	2 (4)	1 (<1)
	Unknown	18 (7)	6 (11)	12 (6)
Gender				
	Female	202 (79)	42 (78)	160 (79)
	Male	47 (18)	5 (9)	42 (21)
	Unknown	8 (3)	7 (13)	1 (<1)
Age				
	<65	45 (18)	5 (9)	40 (20)
	66–75	87 (34)	15 (28)	72 (35)
	76–85	81 (31)	12 (22)	69 (34)
	>85	38 (15)	18 (33) **	20 (10)
	Unknown	6 (2)	4 (7)	2 (1)
Marital Status				
	Married	57 (22)	8 (15)	49 (24)
	Single	58 (23)	13 (24)	45 (22)
	Divorced	56 (22)	4 (7)	52 (26)
	Widowed	71 (28)	23 (43)	48 (24)
	Unknown	15 (5)	6 (11)	9 (4)

* 630 education packages were distributed during the project; 54 ‘in person’ attendees completed the survey; 203 of 576 who received materials for ‘at home’ use completed and returned a survey, for a response rate of 35%. Twenty-one percent (21%) of surveys were from ‘in-person attendees’; 79% from ‘at-home’. ** ‘in-person’ learners were more likely to be Black (*p* < 0.001) and over age 85 (*p* = 0.009) compared to ‘at-home’ learners.

## Data Availability

The data presented in this study are available on request from the corresponding author. Preliminary data was presented during Student Poster Presentation at American Geriatrics Society Conference Spring 2021.

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
