# Peer review of "Targeted Telehealth Education Increases Interest in Using Telehealth among a Diverse Group of Low-Income Older Adults"

_ijerph, 2022, doi:10.3390/ijerph192013349_

Round 1

Reviewer 1 Report

The manuscript has been well edited and shows the main aspects of the work clearly and clearly. However, the last two sentences could be formulated more clearly in the abstract and the conclusion could be supplemented by one or two other important aspects in the flow test.

Reviewer 2 Report

Telehealth has proven to be convenient particularly during the pandemic. I appreciated the assessment of Telehealth use after education. Below are some concerns and comments:

1. Although the study uses post-survey results to assess education, it would have been nice to see measurements of actual Telehealth usage by these particular individuals. Would the authors be able to gather that data?

2. Some of the results are questionable due to the unequal representation of at-home and in-person education. This is also true for sex (more females), race, and age. The authors claim that in-person education was more effective in interest in future Telehealth usage; however, this statement is questionable due to the much higher at-home participants vs in-person, n=203 and n=54 respectively. 

3. Also, there were significant differences found between in-person and at-home learners (race and age). Does this affect the results?  

Reviewer 3 Report

1. The discussion section suggests that some of the content can be dialogued with Introduction. for example One significant barrier to video visits is reliable internet access, which is a significant barrier for older adults in the U.S.

2. In Materials and Methods, please explain the exact number of cases received and whether there is any loss.

3. Blacks were more likely to request more information in ‘in-person’ and ‘at-home’. Please explain if it is related to resource acquisition capability and resource allocation.

Reviewer 4 Report

First, congratulations to the authors for their effort and interest in this relevant topic. The following are my observations.

 Abstract

-         Provide greater specificity in the statement of the purpose of the study.

-         Indicate the methodological design of the study.

1.- Introduction

-         It is recommended to provide more background on the results of other telerehabilitation intervention experiences.

-         Provide more specificity in the statement of the purpose of the study and align with the abstract.

2.- Materials and Methods

-         State the methodological design and type of study.

-         Provide further background and description of the intervention carried out.

-         Indicate the process (criteria) of elaboration of the questionnaires.

-         Apparently typing error ( , , >,>85; ) (line 114)

-         Explain why the study is exempt from IRB. In addition, the consent of the participants is not stated.

3.- Results

-         Standardize the style (uppercase - lowercase) of "P" (line 126)

4.- Discussion.

-         Provide further background to argue "Studies demonstrate that in-person education can identify and address telehealth barriers, and help older adults overcome these" (line 202).

-         It is suggested that Figure 1 be associated with results.

-         Provide greater specificity of the results obtained according to countries (indicate countries) (line 216).

-         Provide greater methodological limitations of the study associated with intervention studies.

Round 2

Reviewer 4 Report

Congratulations for the effort and good work on the suggested improvements.

Best regards